# Legionnaires’ Disease Cases at a Large Community Hospital—Common and Underdiagnosed

**DOI:** 10.3390/ijerph17010332

**Published:** 2020-01-03

**Authors:** Jamie Spiegelman, Theresa Pedutem, Mary June Francisco

**Affiliations:** Humber River Hospital, Toronto, ON M3M 0B2, Canada; TPedutem@hrh.ca (T.P.); MFrancisco@hrh.ca (M.J.F.)

**Keywords:** Legionnaires’ disease, *Legionella*, *Legionella* urine antigen test, pneumonia

## Abstract

Legionnaires’ disease (LD) is a severe pneumonia with a mortality rate of about 10%. The illness remains largely underdiagnosed with outbreaks occurring with alarming incidence. In this study, we assessed the frequency of Legionnaires’ disease among pneumonia cases treated at a large community hospital over a summer season. We invited all admitted patients diagnosed with pneumonia, able to provide a urine sample for an antigen test, presenting from May to October 2018, to enroll in our study; 35 patients were tested for the presence of *Legionella*. Out of 33 patients tested, 9 (28%) were positive for *Legionella*. Three sets of the 9 Legionella cases exhibited spatiotemporal clustering indicative of LD outbreaks. Only one of the 9 *Legionella* UAT-positive patients presented a sporadic case of LD. The number of pneumonia cases in our community confirmed to be LD was strikingly high (28%), compared to other survey studies that report between 3.7% and 14%. These results are consistent with previous knowledge that LD is underdiagnosed and support that routine testing should be considered for all possible LD cases, particularly in the summer months. Such testing is likely to prevent further cases of community acquired LD.

## 1. Introduction

One of the major causes of community-acquired pneumonia (CAP), *Legionella* bacteria are ubiquitous in nature, present in soil and water, and can thrive in man-made water systems, including domestic hot water tanks, decorative fountains, and HVAC (heating, ventilation, air conditioning) cooling towers. These bacteria get their name from a 1976 outbreak in a Philadelphia hotel hosting a meeting of the American Legion. In this largest ever community-associated outbreak of Legionnaires’ disease (LD) in the US, 182 individuals were affected, 29 of whom succumbed to the illness [1]. Despite over four decades since the identification of *Legionella* and its pervasiveness, the bacteria remain a significant threat to human health. The mortality rate of LD, a severe pneumonia for which there is no vaccine, is about 10%.

*Legionella* infection most commonly occurs by inspiration of water droplets containing the bacteria. Water containing *Legionella* can become aerosolized from the numerous water sources in which the bacteria have been permitted to replicate unchecked, perhaps most notably, the cooling towers of buildings greater than 4600 m^2^ [2]. Importantly, prevention of LD is largely possible with mitigation of *Legionella* colonization of man-made water systems. In order to affect the environmental presence of the bacteria, monitoring is essential. Assessment of the bacterial load of water systems, and appropriate mitigation, can improve public safety, but an understanding of the epidemiology of LD cases and outbreaks is also key in preventing additional or repeated exposure to the pathogenic bacteria.

Public Health Ontario (PHO) confirmed 203 cases of Legionnaires’ in the province in 2017 (https://www.publichealthontario.ca), however it is estimated that the number of confirmed cases represents approximately 5% of the cases [2]—the actual number of cases is projected to be upwards of 7000 [3], costing tens of millions of dollars [4]. Since the severity of LD is generally greater than other types of CAP, often requiring treatment in the ICU, the actual cost of LD is likely to be considerably greater. In the US, it is estimated that a single case of LD exceeds $34,000 [5].

Intrigued by the general consensus that LD is underdiagnosed and is more common in the summer months [6], we examined the frequency of LD in the cases of pneumonia hospitalizations at a community hospital in Toronto, ON, Canada, in summer, 2018.

## 2. Materials and Methods

### 2.1. Subjects

All patients presenting with pneumonia, confirmed by X-ray, to the Humber River Hospital Emergency department from May 2018 to October 2018 were invited to participate in this observational and survey study. All procedures performed in studies involving human participants were in accordance with the ethical standards of Humber River Hospital and with the 1964 Helsinki declaration and its later amendments or comparable ethical standards. We obtained REB (Research Ethics Board) approval via Humber River Hospital. Informed consent was obtained from all individual participants included in the study. An information sheet and consent form were provided to participants, who were then assessed for eligibility based on the following inclusion/exclusion criteria:

#### 2.1.1. Inclusion Criteria

Age ≥ 18 years at enrollment.Diagnosis of pneumonia.Come to the Emergency department and requiring admission to our hospital via our Internal Medicine (general medical ward) or ICU (Intensive Care Unit) service.

#### 2.1.2. Exclusion Criterion

Unable to provide urine samples (e.g., dialysis patients).

The following participant details were collected from all subjects:Name, date of birth, contact information, home address, work address, occupation, and travel history if applicable.Comorbidities as documented on the patient chart.Laboratory and imaging information, including final results of *Legionella* testing.Outcome (if known, including death, prolonged hospitalization, discharged [d/c] without complications).

### 2.2. Sample Collection and Assessment of Legionella Infection

All enrolled subjects provided urine samples for the urine antigen test (UAT) which was done at Toronto Public Health. The UAT used at Toronto Public Health the Binax test which has a >70% sensitivity and >99% specific and tests for Legionella pneumophila serogroup 1 only (https://www.publichealthontario.ca).

## 3. Results

From May to October 2018, 35 eligible patients with pneumonia at Humber River Hospital consented to enroll in our study. Of those 35, three were unable to continue in the study. Out of the remaining 32, nine cases (28%) tested positive for Legionellosis (Table 1). Four of the patients required admission to the ICU. Of those admitted to the ICU, three were currently non-smokers, two had never smoked (smoking is a risk factor in contracting LD). Overall, two-thirds of the patients were non-smokers, with 22% active smokers and 11% ex-smokers. The average age of the patients with LD was 68 years old, with one third of them female.

We observed no trends with respect to occupation, however there was a collection of cases in our region. The CDC’s (Center for Disease Control) definition of a Legionnaires’ disease outbreak is an occurrence of two or more individuals contracting LD within 14 days of one another in the same area. Importantly, the plume of *Legionella*-contaminated water vapor from a cooling tower can spread over a radius of greater than 10 km [7]. By these criteria, at least three outbreaks occurred in North York, ON in the summer of 2018 (Figure 1). The first of these arose in July, at which time three individuals residing in a 5.7 km radius contracted LD within 8 days. In August, two patients separated by 6.9 km presented with LD within 2 days, while September saw the most serious outbreak with three patients diagnosed with LD within a week, two of whom required treatment in the ICU.

## 4. Discussion

We used a UAT to assess whether cases of pneumonia were caused by *Legionella*. While it is estimated that 97% of clinical diagnoses are obtained using a UAT [7], these tests only recognize *Legionella pneumophila* serogroup 1 antigens, which accounts for 50%–80% of LD cases [5]. Furthermore, the sensitivity of the test ranges from approximately 60%–100% [5,8]. Thus, it is likely that there were more cases of LD in our community in the summer of 2018 than we report here. While there is no consensus on the overall frequency of LD, it is clear that the disease is underdiagnosed, as most cases of CAP are treated empirically with macrolide antibiotics without pro- or retrospective analysis into disease etiology. [9] It is not necessary to have a test that differentiates beyond the bacterial genus since all *Legionella* species are sensitive to commonly prescribed antibiotics which are generally effective and recommended for both community- and hospital-acquired infections [6]. However, it can be important to differentiate *Legionella* pneumonias from those of other etiology, e.g., *Streptococcus pneumoniae*, where the empiric antibiotic of choice may not affect both genera, such as where beta-lactam agents are used to treat ambulatory pneumonia [10]. Most importantly, in order to mitigate further exposure to *Legionella* from an environmental source of the pathogen, it is necessary to take an epidemiological approach to pneumonia cases. Our study suggests that this is especially relevant in the summer months and is in line with the suggestion UAT be performed for immediate diagnosis but culture be performed for all suspected cases of LD [6].

Those at the highest risk of infection by *Legionella* include smokers, people over the age of 50, and immunocompromised individuals [10]. One of the major sources of *Legionella* exposure is the air conditioning systems of buildings over 4600 m^2^. Heat from such a building is transferred to a cooling tower and then to the atmosphere through the evaporation of water. If the cooling tower becomes contaminated with *Legionella*, bacteria-laden water vapor may be inhaled by the building’s occupants, people in surrounding buildings, and by individuals at street level. *Legionella* from a contaminated cooling tower can be recovered as far as 10 km away from the source [7,11]. Somewhat alarmingly, *Legionella pneumophila* was found in 14 of 20 (70%) US hospitals surveyed across 13 states [12] and 60% of Paris hospital water systems [13]. Additionally, a survey of 196 cooling towers across the US revealed pathogenic *Legionella* strains in 53 towers (27%) [14]. In cooling towers, *Legionella* can grow to outbreak levels within 7 days [15,16].

With cooling towers as a major source of pathogenic *Legionella*, it stands to reason that increased use of HVAC systems in the warmer months would result in a greater number of LD cases. Indeed, this seasonal trend in LD cases has been observed in both the US [17] and Europe [18]. However, it is important to note that the proportion of community-acquired pneumonia cases that is Legionellosis remains somewhat enigmatic, ranging from below 2% to 14% [10]. A multi-center, Canada-wide study reported in 2003, found 3.2% of CAP patients had LD, however that study enrolled patients from January to October, so it is possible that the LD cases, which my constitute a large proportion of summertime pneumonia cases, was diluted by the seasonal spike of pneumonia arising from communicable diseases encountered in the winter.

While our study identified three outbreaks of LD, attention to sporadic cases must not be overlooked. Sporadic cases are common and, since the disease is rare in healthy individuals, may still be indicative of an environmental point source that could initiate additional cases in the future. Further, *Legionella*, while fastidious in laboratory culture, have adapted to survive and replicate in harsh environments, quickly recolonizing man-made water networks even after rounds of heat-shock or biocide treatment.

Understanding the frequency of *Legionella* infection will enable Public Health to reduce exposure to *Legionella* and incidence of LD in the community. While further study with a larger cohort of cases may be warranted, based on the frequency of outbreaks and the high proportion of LD cases in our population, we believe that both UAT and culture testing of all summertime cases of pneumonia for Legionellosis will significantly benefit community health.

## 5. Conclusions

Our study, conducted in the summer of 2018, found that 28% of the pneumonia cases at Humber River Hospital in North York (Toronto), ON were Legionnaires’ disease. Further, most of the cases presented as the result of an outbreak, which is to say that two or more cases from the same area occurred within a couple weeks of one another.

## Figures and Tables

**Figure 1 ijerph-17-00332-f001:**
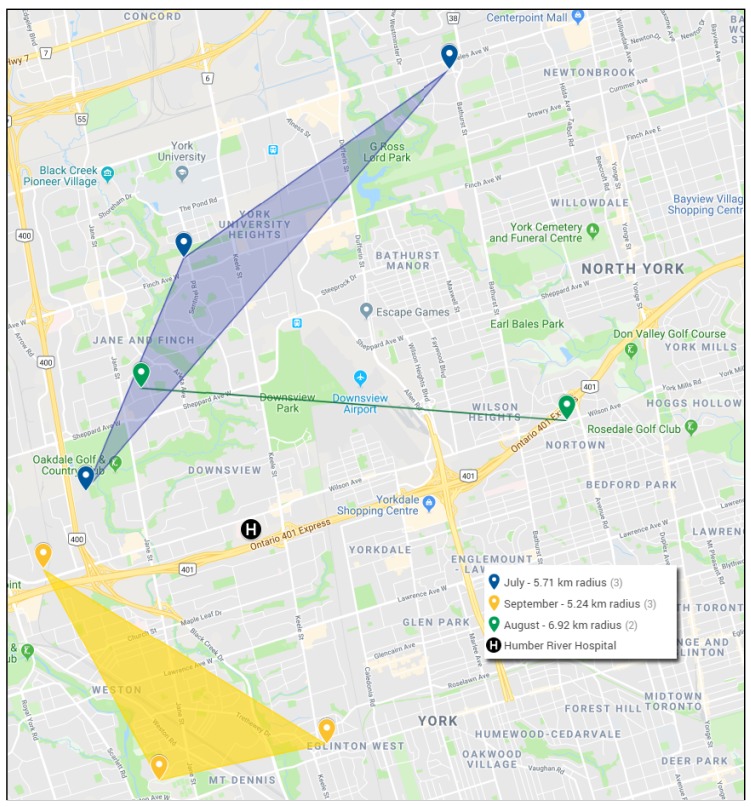
Legionnaires’ disease outbreaks in North York, ON in summer, 2018. Colored pins represent the patients’ residential addresses.

**Table 1 ijerph-17-00332-t001:** Enrolled patient characteristics and LD diagnosis.

Date Collected (Year-Month-Day)	Sex	Age	Home Structure	Occupation/Industry	Smoking History	Co-Morbidities *	Legionella Status	Outcome
2018-07-09	M	57	Detached-house	Driver	Current, 1 pk/d since 1974	HTN	+	ICU admission
2018-07-02	F	87	High-rise building	Retired	Never	CAD, PVD, CKD	+	Prolonged admission
2018-07-06	F	63	High-rise building	Personal support worker	Never	HTN, Asthma	+	D/C without complication
2018-08-21	F	86	High-rise building	Retired Nurse	Never	HTN, RA, AF	+		D/C without complication
2018-08-22	M	71	Semi-detached house	Trucking Industry	Never	HTN, DM	+		D/C without complication
2018-09-03	M	54	Detached house	Automotive Industry	Current, ½ pk/d since 1985	-	+		D/C without complication
2018-09-06	M	65	Detached house	Welder/bus driver	Former, 1 pk/d for 20 y	-	+		ICU admission
2018-09-02	M	69	Semi-detached house	Heating industry	Never	HTN, CAD, Gout	+		ICU admission
2018-09-09	M	64	High-rise building	Insurance	Never	Alcohol abuse	+		ICU admission
2018-08-27	F	36	Low-rise building	Cashier	Current, 1 pk/d since 1998	Asthma, DM	-		D/C without complication
2018-06-27	F	61	Semi-detached house	On disability	Never	Multiple myeloma	-		Death
Not performed	F	86	Detached house	Housewife	Never	HTN	NA		Prolonged Admission
2018-06-06	F	79	Detached house	retired	Never	HTN, CKD, Gout	-		D/C without complication
2018-07-18	F	40	High-rise building	Interpreter	Never	Crohn’s	-		D/C without complication
2018-07-26	M	84	Detached house	Construction worker	Former, 40 y until 1998	COPD, HTN, DM	-		D/C without complication
2018-07-25	F	47	Low-rise building	Housewife	Never	CAD, Grave’s	-		D/C without complication
Not performed	M	88	Semi-detached-house	Business owner	Former, 40 y until 1995	HTN, lung cancer, DM	NA		D/C without complication
2018-07-25	M	74	High-rise building	Retired, government	Former, 50 y until 2013	Lung cancer, DM	-		D/C without complication
2018-08-08	M	87	Detached house	Retired Librarian	Former, 56 y until 2003	HTN, COPD, CKD	-		D/C without complication
2018-08-05	F	90	High-rise building	Housewife	Current, ½ pk/d since 1958	CAD, AF, CKD	-		D/C without complication
Test cancelled	M	46	Townhouse	Construction	Current, ½ pk/d since 1998	CHF	NA		D/C without complication
2018-08-02	M	53	Semi-detached house	Construction worker	Current, 5 cig/wk since 1985	CAD	-		D/C without complication
2018-08-12	M	81	Detached house	Machinery Inspector	Never	CAD, DM	-		D/C without complication
2018-08-10	M	60	High-rise building	Factory worker	Former, 30 y until 2014	CHF, DM, HTN	-		D/C without complication
2018-08-20	M	74	Low-rise building	Rental car/Airport	Former, 46 y until 2000	HTN	-		D/C without complication
2018-08-19	F	93	Detached house	Retired	Never	HTN	-		D/C without complication
2018-08-30	M	77	Semi-detached house	Stone Mason	Former, 5 cig/d until August 2018	AF, HTN, gout	-		D/C without complication
2018-08-28	M	66	High-rise building	Truck driver	Never	-	-		D/C without complication
2018-08-30	M	88	Detached house	Retired	Never	CAD, Prostate cancer	-		Prolonged Admission
2018-09-05	M	58	Detached house	Maintenance	Current, 1 pk/d since 1978	COPD	-		Prolonged Admission
2018-09-04	M	71	Detached house	Car industry	Former, 20 y, 2–3 pk/d until 1999	CAD, PVD, CKD, DM	-		Prolonged Admission
2018-09-02	M	57	Detached house	Accounting	Former, 5 cig/d until 1984	HTN	-	D/C without complication
2018-09-11	M	75	Detached house	Housewife	Never	CHF, AF	-	Prolonged Admission
2018-09-19	M	85	Detached house	Business/machine shop	Never	AF, CAD	-	ICU admission
2018-09-17	M	32	Detached house	Security guard	Never	Asthma, Obese	-	D/C without complication

* Definitions: HTN = hypertension, CAD = coronary artery disease, COPD = chronic obstructive pulmonary disease, CKD = chronic kidney disease, PVD = peripheral vascular disease, RA = rheumatoid arthritis, AF= atrial fibrillation, DM = diabetes mellitus, CHF = congestive heart failure.

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
