# Peer review of "Legionnaires’ Disease Cases at a Large Community Hospital—Common and Underdiagnosed"

_ijerph, 2020, doi:10.3390/ijerph17010332_

Round 1
Reviewer 1 Report
International Journal of Environmental Research and Public Health
Legionnaires’ disease cases at a large community hospital – common and underdiagnosed.
Overall: This work presents findings from a small clinical survey of individuals diagnosed with pneumonia at a large community hospital during summer months. Participants were given a urinary antigen test (UAT) to assess how many of the participants would test positive for Legionnaire’s disease. This work is valuable but additional description of methods and limitations will be necessary to strengthen this work. All major comments should be addresses before accepting for publication.
Comments:
Minor:
Abstract: (Editorial comment) Page 1, Line 18. Sentence starting: “Our results add rigour to claims that LD is underdiagnosed and support that routine testing and culture should be considered for all possible LD cases, …” I think the term “that” needs to be defined or sentence needs to be reworded. Introduction: Page 1, Line 35: “…towers of buildings greater than 4600m2” Requires a reference/citation to support this. Introduction: Last sentence of the section: “We found that 28% of patients hospitalized for pneumonia had LD.” This should be removed since it belongs in the results, discussion or conclusion section. Materials and Methods: #3 for Inclusion Criteria: “via our Internal Medicine or ICU service,” What is “Internal Medicine”? Is there a different terminology you can use or clarification you can add? Materials and Methods: #4 for participant details: “Outcome (if known)” – can you clarify what “Outcome” stands for? Results: Lines 87-89: “Four of the patients required admissions …three were non-smokers, two had never smoked...” These numbers don’t add up, probably a typo. Further I would move this sentence below the description of your overall participation characteristics – as it reads out of place. Results: Lines 91-92: Sentence “…., however a spatiotemporal clustering of cases was observed.” This part of the sentence should be moved to the start of the very next paragraph. Table 1. Could be cleaned up a bit: What does “PSW” stand for? Smoking history is missing number of years of smoking for some smokers. Need footnotes to explain any acronyms in the paper Table 1. Why aren’t comorbidities included? Discussion – be thoughtful about use of legionellosis (and not LD) as you have not previously defined this. Discussion, paragraph 2 – “Those at highest risk of infection by Legionella include smokers..” This should be moved to the introduction because this background would be useful for supporting your selection of Participant details.Major:
Materials and Methods: Can you describe how many individuals met inclusion criteria but did not participate? Do you have any additional information about these individuals and how they may or may not compare with your participants?Can you also describe how your hospital represents the larger community – would one assume that all cases of pneumonia would come to this hospital? How does the exclusion of dialysis patients affect representativeness? Would other possible cases go to other hospitals or specialists?
This will help us better understand how your participants are representative of the hospital/community population.
Need to describe “spatiotemporal clustering” methodology in the Methods section. However, I don’t think your methods are strong enough to include this analysis in this paper. Two standard options for identifying a community cluster: 1. To identify a community cluster you link the cases with a common exposure – not just in time, particularly given that your clusters span relatively wide locational areas, but you provide no evidence of a shared source. This CDC definition is largely used for identifying facilities outbreaks such as hotels or long-term care. 2. You could identify a community cluster based on an overall increase from baseline but it does not appear that there were any methods used to statistical assess “increases” in disease. In the Methods you need to describe what you are going to do with the participant details you collected. Why did you collect them, did you analyze these, how were they analyzed? Results: “We observed no trends with respect to occupation…” This is related to the comment above as well – what methods were used to reach this conclusion, but further when I reviewed Table 1, I found the occupation very interesting, particularly that there may be 5 or more occupations related to driving – which is a known high-risk industry for LD.
Author Response
Dear Reviewer 1 -
Here are my point form comments to your review:
Page 1, Line 18: The sentence “Our results add rigour to claims that LD is underdiagnosed and support that routine testing and culture should be considered for all possible LD cases, …” has been changed to (to clarify it): "These results are consistent with previous knowledge that LD is underdiagnosed and support that routine testing and culture should be considered for all possible LD cases, particularly in the summer months." Page 1, Line 35: reference added Last sentence of the section: “We found that 28% of patients hospitalized for pneumonia had LD.” This should be removed since it belongs in the results, discussion or conclusion section. - agreed - this has been deleted. Materials and Methods: #3 for Inclusion Criteria: “via our Internal Medicine or ICU service,” What is “Internal Medicine”? Is there a different terminology you can use or clarification you can add? - Internal Medicine or ICU are generally accepted terms in a hospital setting. For completeness here, I clarified this here in brackets next to internal medicine to include "general medical ward" and I expanded ICU as Intensive Care Unit. Materials and Methods: #4 for participant details: “Outcome (if known)” – can you clarify what “Outcome” stands for? - I have defined common outcomes from hospitalized LD patients in brackets ie including death, prolonged hospitalization, discharged [d/c] without complications Results: Lines 87-89: “Four of the patients required admissions …three were non-smokers, two had never smoked...” These numbers don’t add up, probably a typo - No, this is correct - 4 patients were admitted to the ICU and out of the 4 admitted patients to the ICU there were no currently smoking and 2 never smoked - I have clarified this with the word current. Results: Lines 91-92: Sentence “…., however a spatiotemporal clustering of cases was observed.” This part of the sentence should be moved to the start of the very next paragraph. - agreed - this has been done. Table 1. Could be cleaned up a bit: What does “PSW” stand for? Smoking history is missing number of years of smoking for some smokers. Need footnotes to explain any acronyms in the paper - we included data we had on the patients/ short forms have been defined. Table 1. Why aren’t comorbidities included? - agreed this is useful, I have added it to the table. Materials and Methods: Can you describe how many individuals met inclusion criteria but did not participate? Do you have any additional information about these individuals and how they may or may not compare with your participants? - 35 total patients met inclusion and 3 were unable to be enrolled in the study. Can you also describe how your hospital represents the larger community – would one assume that all cases of pneumonia would come to this hospital? How does the exclusion of dialysis patients affect representativeness? Would other possible cases go to other hospitals or specialists? - the hospital is a standard community hospital in the Toronto area. Not all pneumonia comes to hospitals, some go to their family doctor and treated as an outpatient. we are unable to enroll dialysis patient as they don't make urine and therefore unable to test them using the urine test. -for “spatiotemporal clustering” - we believe the trend we saw represents an area defined by the CDC in our population.
Reviewer 2 Report
The manuscript “Legionnaires’ disease cases at a large community hospital-common and underdiagnosed” is well written. The figures and tables are well presented.
I have a question: the used methodology aims to identify cases of LD by Legionella pneumophila serogroup 1. Why the authors did not use other methods to detection and identify other Legionella species in patients negative for the Binax test? Moreover, the introduction section is poorly documentated by literature.
There are few comments and major corrections that should be solved before this manuscript can be considered for publication.
INTRODUCTION
Line 24-40: the authors are requested to expand the list of references (e.g. about risk factor, i.e. alcool, smoke, gender, age etc).
Line 26 and 45: Please specify in full the acronyms (HVAC and ICU)
RESULTS
Line 95: Please specify in full the acronym "CDC".
DISCUSSION:
Line 120: please add ß to -lactam agents. Probably it is a error caused by the conversion from word to PDF. Possible?
Author Response
Here are my replies to your comments in point form:
the used methodology aims to identify cases of LD by Legionella pneumophila serogroup 1. Why the authors did not use other methods to detection and identify other Legionella species in patients negative for the Binax test? - this is the only test available in the Canada (tested by Public Health) - the only other method is to do a bronchoscopy on all patient and then a DNA test on the sputum which is not standard of care. This would be almost impossible to do. The urine test is the only standard of care Moreover, the introduction section is poorly documentated by literature. - this has been updated in the manuscript Line 26 and 45: Please specify in full the acronyms (HVAC and ICU) - updated in manuscript Line 95: Please specify in full the acronym "CDC". - updated Line 120: please add ß to -lactam agents. Probably it is a error caused by the conversion from word to PDF. Possible? - updatedThank you for your kind review!
Reviewer 3 Report
The authors assess the frequency of Legionnaires’ disease (LD) among pneumonia cases treated at a large community hospital from May to October 2018. Thirty – five patients able to provide an urine sample for an antigen test have been enrolled in this study. Unfortunately, the study does not have a clear research question and anyway is not designed appropriately to answer that question. It is universally known that LD is underdiagnosed but the protocol of this study does not appear useful for studying this topic.
The frequency of LD is calculated on the number of patients with pneumonia able to provide an urine sample without considering the number of pneumonia excluded from the study because unable to provide urine samples, so this finding is misrepresented. Moreover, no data about the surveillance in other years or in other months of the year 2018 besides the summer was reported. The high incidence of LD reported (28%) could be simply due to an epidemic event, however the authors do not provide any information about the incidence of LD in that area and therefore no conclusions can be drawn.
The authors state the presence of three different outbreaks without supporting this statement with an epidemiological survey or environmental sampling in order to define the source of infection. Moreover, there is no trace of survey on clinical strains, as cultural assay and molecular genotyping. These investigations are essential to identify the same Legionella strains in different patient and therefore to suppose the presence of an outbreak.
In the manuscript the definition of outbreaks is based only on arbitrary spatial and temporal criteria (see lines 95-102 and 159-160). To date, the CDC indicates to extend the temporal criterion up to 12 months; for this reason, the period of 14 days reported by the authors seems not appropriate. The spatial criterion is based only on the radius of legionellae spread of the cooling towers but the authors do not specify whether there were cooling towers in that area. It cannot be excluded that there were one or more sources of infection other than the cooling towers as the municipal water system.
Author Response
Dear Reviewer -
Thanks you for your honest review. We have reviewed your concerns and have corrected the manuscript as best we can, as follows:
we believe the purpose of this study was to show that LD is underdiagnosed which supports previous literature and this was the main research question posed. we do mention in our methods that if a patient was unable to provide a urine sample ie dialysis patients, we excluded this patients as they cannot be tested using current diagnostic tests for LD - unfortunately the diagnostic tests universally used (ie the urine Ag test), is really the standard of care but is not very good. The only other method to diagnose LD is to get a sputum sample usually using bronchoscopy which is not a reasonable methods in a study like ours and is not the standard of care. we do not have data on other years or surveillance data for our population as it does not really exist. In our introduction we note this and the public health Ontario follows it in a broader population. the purpose of this study was not do look at molecular strains/genotyping etc, this would be a much different and complex study which we didn't present here. in term of spatial-temporal data, we presented the data and trends that we had during the study as a representation and sample size of a broader problem. There are many cooling towers on every apartment building in the area and impossible to define in this study. Hence we on purpose have not pinpointed the exact source.
thank you!
Round 2
Reviewer 1 Report
There were two ‘major comments’ submitted with my first review that were not addressed by authors:
In the Methods you need to describe what you are going to do with the participant details you collected. Why did you collect them, did you analyze these, how were they analyzed? Results: “We observed no trends with respect to occupation…” What methods were used to reach this conclusion, but further when I reviewed Table 1, I found the occupation very interesting, particularly that there may be 5 or more occupations related to driving – which is a known high-risk industry for LD.Among the “addressed” comments the responses and subsequent revisions were inadequate:
Materials and Methods: Can you describe how many individuals met inclusion criteria but did not participate? Do you have any additional information about these individuals and how they may or may not compare with your participants? Can you also describe how your hospital represents the larger community – would one assume that all cases of pneumonia would come to this hospital? How does the exclusion of dialysis patients affect representativeness? Would other possible cases go to other hospitals or specialists? This will help us better understand how your participants are representative of the hospital/community population.Although authors address this, they made no meaningful addition to the paper to highlight to other readers the representativeness of their hospital study to the larger community.
Comments regarding author analysis and conclusions regarding spatiotemportal clustering are not adequately addressed. Specifically, methods are not described in the Methods section as they should be. And more importantly, the identified clusters are linked in “time” only and not “area” as stated by the authors. As there is no epidemiologic data to suggest that cases had a common exposure source, the conclusion that they are “clustered” because their residential addresses are within a large community are as many as 7km apart is not sufficient. The CDC definition is intended to define cases that share a common exposure source and “location” such as a facility. On the other hand community clusters are identified largely through statistical means comparing a baseline disease exposure to the current disease burden to show a statistical increase or anomaly – this was not performed by the authors. As it stands, the results they present do not meet analytic standards.Additional comments:
Scope of the discussion is unclear. Authors suggest that largely CTs are the source of exposure and disease which is too narrow and there is a lot of literature surrounding sources of Legionella exposure that are unacknowledged by the authors. In the second paragraph of the Introduction authors use the word “inspiration” of water droplets. I am not familiar with this terminology – should use “inhalation or aspiration” The term legionellosis is used but had not been defined in the introduction. Does the obtained consent from patients include permission to publish their residential address?Author Response
Dear Reviewer -
Here are my comments/ revisions to your latest concerns:
"Can you describe how many individuals met inclusion criteria but did not participate? Do you have any additional information about these individuals and how they may or may not compare with your participants?comments: this is commented on line 86-88. Out of 35 patients, 3 were not able to participate. Reasons were mostly due to lack of access or discharge. Can you also describe how your hospital represents the larger community – would one assume that all cases of pneumonia would come to this hospital?
Toronto has about 10 community hospitals, so not all cases of pneumonia who need to be hospitalized comes to our hospital.
How does the exclusion of dialysis patients affect representativeness?
We cannot comment on this as we do not have data or info on this.
Would other possible cases go to other hospitals or specialists?
Other cases may be treated without coming to hospital or going to other hospitals in the area. As for the other comments - we believe we adequately corrected these or defined them in the manuscript. The consent did include where patients live.
Reviewer 3 Report
I have seen the changes that the authors have made in the text and I have taken note of their objective impossibility to supply further information on some aspects that I believe are fundamental to satisfy the objective of the research. Despite the changes, the manuscript remains unsuitable for publication.
Author Response
Dear reviewer:
Than you for you comments. However, we don't agree with you. We believe the data presented gives the medical as well as public health more data on how underdiagnosed LD is in our community.
Thanks you.